# Taxon-specific redox conditions control fossilisation pathways

Nora Corthésy ✉, Jonathan B. Antcliffe & Farid Saleh ✉

The preservation of fossils in the rock record depends on complex redox processes. Redox conditions around different decaying organisms have rarely been monitored in the context of experimental taphonomy. Here, micro-sensors were used to measure redox changes around decomposing carcasses of various taxa, including shrimp, snail, starfish, and planarian. Our results show that different decaying taxa lead to various post-mortem environmental redox conditions. Large carcasses tend to reach reducing conditions more rapidly than smaller ones. However, size does not explain all observed patterns, as environmental redox conditions are also influenced by the nature of the organic material. For instance, taxa with higher proteins-to-lipids and (proteins + carbohydrates)-to-lipids ratios tend to achieve reducing conditions more rapidly than others. The generation of distinct redox environments around different taxa originally put under the same original environmental conditions suggests that various fossilisation patterns of macrofossils and molecules can co-occur within a single sedimentary layer.

Numerous lines of evidence enable us to reconstruct past life on Earth using data from the fossil record. These include body fossils, trace fossils such as burrows and footprints, and chemical fossils, which consist of molecules and biomarkers left behind by organisms in the rock record. Both body and chemical fossils undergo similar tapho-nomic processes, including bacterial decay during early diagenesis, maturation during late diagenesis and metamorphism, and modern weathering[1–3]. These processes can lead to either information loss due to the degradation of specific molecules and morphologies, or infor-mation retention when environmental conditions stabilise certain chemistries or anatomies[3,4].

Redox conditions are a crucial factor that determines what is lost or retained in the rock record. Redox condition refers to the overall oxidation-reduction state of an environment, which is determined by the balance between oxidising and reducing processes. It is influenced by the presence and activity of electron donors (reducing agents) and electron acceptors (oxidising agents). The redox condition of an environment can be deciphered by investigating the redox potential or oxidation-reduction potential (ORP) of this environment. The redox potential is a measure of the tendency of a chemical species to gain or lose electrons in a redox reaction[5]. It is typically expressed in millivolts (mV) and indicates whether a system is more oxidising (positive ORP)

or reducing (negative ORP)[6]. Generally, when organic matter is exposed to oxidative conditions, it decays more rapidly than it would under reducing conditions[7]. However, this is not always the case, especially since numerous chemical and biological processes are interconnected. For instance, some bacteria recycle organic material under reducing conditions more efficiently than under oxidative conditions[8].

Authigenic mineralisation of organic material is a process by which labile morphological details get replicated by minerals and is a major process for soft tissue preservation or exceptional fossil pre-servation in the rock record[4,9,10]. Authigenic mineralisation also depends on the redox potential[11], since it often occurs under reducing conditions[4,10,12], albeit with some rare exceptions[13]. For instance, pyr-itisation happens under sulphate-reducing conditions and has been shown to replicate in $FeS_2$ numerous types of organisms, including arthropod bodies with neural anatomies[14,15], echinoderms with water vascular systems[16,17], and prokaryotic cells[18]. Phosphatisation happens under reducing phosphorus release conditions, and can replicate soft anatomy in detail in calcium phosphate $[Ca_3(PO_4)_2]$, as observed in muscle tissues[19], trilobite guts[20], and ray-finned fish embryo fossils[9].

Decay experiments conducted under controlled laboratory con-ditions are particularly useful for understanding processes such as the

Institute of Earth Sciences, University of Lausanne, Géopolis, Lausanne, Switzerland. ✉e-mail: nora.corthesy@unil.ch; farid.nassim.saleh@gmail.com

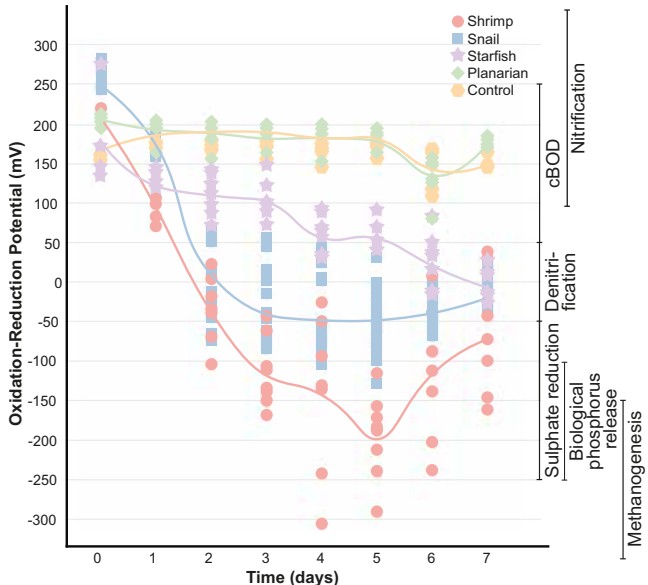

**Fig. 1 | Oxidation-reduction potential (ORP) values (mV) over time for four different animals.** The biochemical reactions corresponding to different redox values: nitrification (+100 to +350 mV), carbonaceous biochemical oxygen demand (cBOD) (+50 to +250 mV), denitrification (−50 to +50 mV), sulphate reduction (−250 to −50 mV), biological phosphorus release (−250 to −100 mV), and methanogenesis (−400 to −175 mV). The shaped points represent individual measurements of redox potential for each animal and control, and the lines follow the average redox potential of each animal and control at each time point. Source data is provided as a Source Data file.

decay and mineralisation of organic matter[8,21–31]. Although redox potential plays an important role in organic matter decay and mineralisation, little work has been done to experimentally investigate what changes in redox conditions occur during organic matter degradation[32–35], and no work has ever investigated how different phyla and organic matter compositions impact redox conditions after death in a single study with directly comparable experimental set ups. This study monitored changes in redox conditions surrounding carcasses of four different animals (shrimp, snail, starfish, and planarian) for a week using microsensors. The results show that redox conditions vary significantly between taxa, depending on the mass and nature of the decaying organic material.

## Results and discussion
### Heterogenous drop in redox potential between taxa
Redox potential values decrease for the four taxa during decay. However, these values are very different from one taxon to another. While the ORP values are similar at the beginning of the experiment (Fig. 1) with no significant differences between them (Table S2), from the second day, the redox potentials are significantly different between the four animals (Fig. 1). The values of shrimps are lower than the values of snails (Contrast analysis of GLM, $p = 0.0140$, $t$-ratio = -3.175, Table S2), which are lower than the values of starfish (Contrast analysis of GLM, $p < 0.0001$, $t$-ratio = −6.973, Table S2) that are lower than the values of planarians (Contrast analysis of GLM, $p = 0.0001$, $t$-ratio = 4.578, Table S2). The significant differences in redox potentials between the animals cannot be explained by the difference in number of samples between snails and the other animals since the variances across ORP values in the animals are very different but this is not due to the larger number of snail samples (Supplementary Materials, Fig. S1 & Table S1).

For shrimps and snails, there is a drastic drop in ORP values between days 1 and 5, at which point they average -200 mV and -50 mV, respectively (Fig. 1). After day 5, the redox values of shrimps and snails

start to increase (Fig. 1). For starfish, ORP values slowly decrease, but there are no drastic drops as for shrimps and snails (Fig. 1). However, after 7 days of decay, the ORP values of starfish are very similar to snails (Fig. 1). For planarians, there is almost no change in ORP values during decay, and the values stay stable until the end of the experiment with a slight decrease then an increase on day 6, in a similar way to the values of the controls in which no organic matter was present (Fig. 1).

ORP ranges are indicative of the following chemical reactions, termed "ORP zones": nitrification or the production of nitrate (+100 to +350 mV), carbonaceous biochemical oxygen demand (cBOD) or the oxidation of carbonaceous material (+50 to +250 mV), denitrification or the reduction of nitrate (−50 to +50 mV), sulphate reduction into sulphides (−250 to −50 mV), biological phosphorus release when the oxygen in phosphate is consumed and phosphorus is released (−250 to −100 mV), and methanogenesis when carbon dioxide becomes the only source of oxygen in the environment and is reduced into methane (−400 to −175 mV)[6]. Generally speaking, the transition from nitrification to methanogenesis corresponds to a decrease in oxygen availability in the environment. For instance, under nitrification conditions, excess oxygen is present and readily reacts with ammonia, transforming it first into nitrite and then into nitrate. Under denitrifying conditions, free oxygen becomes limited, and nitrate serves as the primary oxygen source in the system, being reduced back into nitrite and ammonia. When nitrate is depleted, sulphate and phosphate become the main oxygen sources. Methanogenesis occurs when the only remaining oxygen in the system is in the form of carbon dioxide, which is reduced to methane. In this situation, the measured ORP values are a simplified tool to understand not only the redox potential within an environment but also its underlying chemical and biological processes[36]. Under some of these ORP zones, some mineralisation processes can occur, such as pyritisation under sulphate-reducing conditions and phosphatisation in the phosphorus release zone.

Different animals do not show the same proportions of specimens in the different ORP zones for the duration of the experiment (Fig. 1, S3). Planarians have ORP values corresponding to nitrification and cBOD degradation from day 0 to day 7 (Fig. 1, S3). Starfish do not reach values below −50 mV and most individuals stay in the range of nitrification and cBOD degradation values (Fig. 1, S3), with some of them reaching denitrification from day 4 (Fig. 1, S3). Only shrimps reach methanogenesis values (Fig. 1, S3). Shrimps and snails are the only ones to reach ORP values corresponding to sulphate reduction and phosphorus release (Fig. 1, S3). In this situation, if a source of iron is available, pyritisation, the formation of pyrites, could happen around decaying carcasses of shrimps and snails, where sulphates are reduced into sulphides. Moreover, the reduction of phosphates around shrimp and snail carcasses releases phosphorus into the environment, which can then react with available calcium, allowing the precipitation of calcium phosphate through phosphatisation. Thus, it is possible to suggest that different taxa drive contrasting post-mortem redox conditions, which can lead to various mineralisation patterns.

These observed differences regarding the mineralisation potential of the four taxa are not caused by the ORP variations observed at the start of the experiment (see variations in values at day 0; Fig. 1), as similar trends and significant differences between phyla are observed when ORP values are normalized (Fig. 2A, Table S3). For instance, normalized ORP values for shrimps and snails diminish rapidly compared to other phyla, then slowly increase by day 6 (Fig. 2A as in Fig. 1). Starfish redox potential values slowly decrease until day 5, when a slight increase is observed, followed by a drop until the last day of the experiment (Fig. 2A as in Fig. 1). The daily differences in planarians are almost negligible (Fig. 2A), though there is a slight decrease in ORP values on the sixth day, followed by a slight increase on the seventh day (Fig. 2A as in Fig. 1). The reason for the increase in redox potential after 5 days of decay for almost all organisms (Fig. 1) may be due to a succession of bacterial communities, causing a change in metabolism

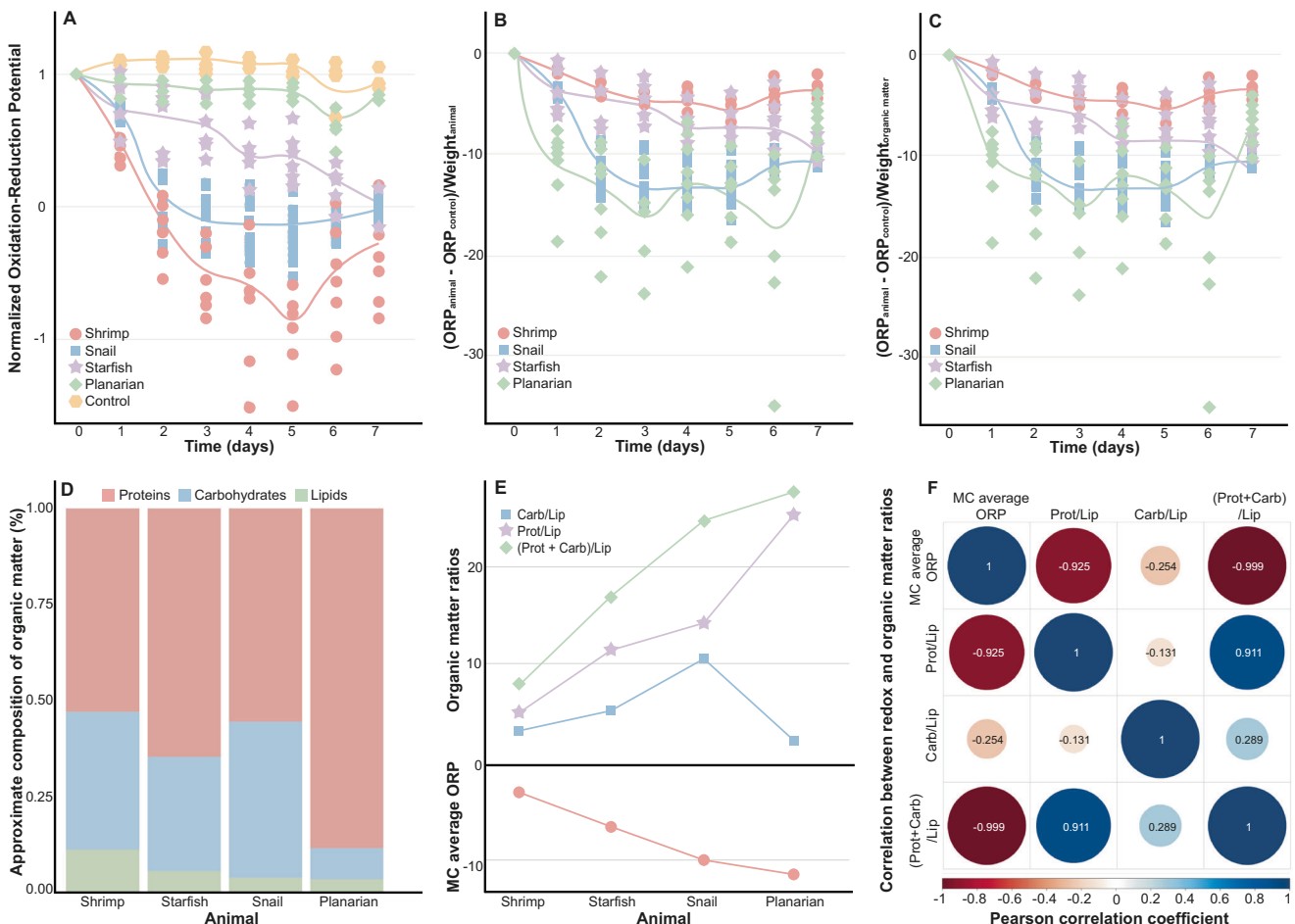

**Fig. 2 | Normalized and body-mass corrected oxidation-reduction potential (ORP) values. A** Normalized ORP values over time for the four different taxa and controls. **B** Redox potential corrected for animal mass: average normalized ORP values of controls subtracted from the normalized ORP values of animals, and divided by the average weight of the animals. Low values reflect a notable drop in ORP values when correcting for the mass of the organism, while zero values reflect no drop in ORP values after mass correction. In other words, showing the lowest values in (**B**) planarians leads to the biggest drop in ORP values followed by snails, starfish, and shrimps of the same size. **C** Redox potential corrected for organic matter mass: average normalized ORP values of controls subtracted from the normalized ORP values of animals, and divided by the average weight of the organic matter by subtracting the mass of biomineralized matter from the mass of animal.

The shaped points in (**A–C**) represent individual measurements and the lines are following the average of each animal at each time point. **D** Proportions of proteins, carbohydrates and lipids in the organic matter of shrimp (*Neocaridina davidi*), starfish (*Asterina sp.*), snail (*Stenophysa marmorata*) and planarian (*Procotyla sp.*). **E** Mass-corrected (MC) average redox potential for each animal, and ratios between the components of organic matter: Carbohydrates/Lipids (Carb/Lip), Proteins/Lipids (Prot/Lip), (Proteins + Carbohydrates)/Lipids (Prot + Carb)/Lip). **F** Pearson correlation coefficients between the mass-corrected (MC) average redox potential for each animal and the ratios between the components of organic matter. The blue colour represents a positive correlation, while the red colour represents a negative correlation (anti-correlation). The larger and darker the circles, the higher the correlation coefficient.

and therefore in redox values. However, more research is needed to understand this trend.

### Redox potential is influenced by the mass and type of decaying organic matter

When the redox potential values are adjusted to account for the mass of the entire animal, new patterns are observed. Snails and planarians exhibit the most notable changes in ORP values relative to their mass, while starfish and especially shrimp show very low changes in ORP over time (Fig. 2B). Between days 2 and 5, the changes in ORP values for planarians and snails are significantly higher than those of starfish and shrimps (Contrast analysis of GLM, $p_{Planarians - Shrimp} < 0.0001$, $t\text{-}ratio_{Pl - Sh} = 8.306$; $p_{Planarians - Starfish} < 0.0001$, $t\text{-}ratio_{Pl - St} = 7.487$; $p_{Shrimp - Snail} < 0.0001$, $t\text{-}ratio_{Sh - Sn} = -7.671$; $p_{Snail - Starfish} < 0.0001$, $t\text{-}ratio_{Sn - St} = 6.701$, Table S4). A similar trend is observed when correcting the data based only on the mass of the organic material (Fig. 2C) and removing the mass of the mineralised parts, which is unlikely to contribute to redox variation due to its decay-resistant mineralogical composition. This indicates that, theoretically, if all four animals had the same weight (and thus the same biomass-to-water ratio in the experiment), planarians and snails would induce more substantial changes in redox conditions, leading to a more marked drop in ORP values compared to both starfish and shrimps. There are probably many causes behind these patterns including but not limited to the shape of the organism (i.e., surface area) and the composition of the respective organic material[37–41] and associated bacteria. In other words, different carcasses can oxidise (i.e., consume oxygen) at different rates during decay[4], driving carcasses into distinct ORP zones.

The organic matter of all four animals contains over 50% protein[37–41]. In the case of planarians, their organic matter is composed almost exclusively of proteins, with a little over 10% consisting of carbohydrates and lipids (Fig. 2D)[41]. Shrimps, snails, and starfish contain more carbohydrates than planarians, with shrimps having the highest proportion of lipids (Fig. 2D)[37–40,42]. These different proportions of organic matter components could explain the variations in oxygen consumption and ORP values. For instance, proteins are known

to decay the fastest, with lipids being the most difficult to degrade[4]. This suggests that planarians, whose organic matter is primarily composed of proteins (Fig. 2D), degrade and consume oxygen faster than the other animals, which could explain why planarians show the most significant differences in ORP values when correcting for size (Fig. 2B, C). In contrast, shrimps are richer in lipids (Fig. 2D), which are harder to break down and consume oxygen at slower rates than the other animal groups. When taking the average redox values corrected by organic matter mass (Fig. 2C) and comparing them to different organic material ratios (Fig. 2E), it becomes clear that both proteins and lipids dictate changes in redox conditions more than carbohydrates. For instance, the decrease in ORP values corrected by organic matter mass is accompanied by an increase in proteins-to-lipids (Pearson test, $r = -0.9251565$, Table S5) and (proteins + carbohydrates)-to-lipids (Pearson test, $r = -0.9990038$, Fig. 2F, Table S5) ratios (Fig. 2E). This essentially means that protein quantities are anti-correlated to mass-corrected ORP values (Fig. 2F), and that decreasing ORP values and achieving reducing conditions is easier with higher protein quantities and lower lipid values. On the contrary, when accounting for carbohydrates on their own (e.g., carbohydrates-to-lipids ratio), no correlation is observed with average redox values corrected by organic matter mass (Fig. 2E, $r = -0.2538133$, Fig. 2F, Table S5), indicating that the role of carbohydrates in influencing redox conditions is ambiguous. These results indicate that organisms with a higher proportion of lipids and a lower proportion of proteins break down and consume oxygen at slower rates than those with a low proportion of lipids and a high proportion of proteins, leading to differences in ORP values (Fig. 2E). This may explain why, when correcting for size differences, shrimps show the least dropdown in ORP values, followed by starfish, snails, and planarians (Fig. 2C, E). However, these results must be nuanced with the size of the organism. Despite planarian organic matter being the fastest to break down and showing the most significant changes in ORP values (Fig. 2B, C), the raw data show that, due to the greater mass of shrimps, vials with shrimp carcasses were the only ones to eventually reach ORP values as low as those characteristics of methanogenesis (Fig. 1, S3). This is mainly because there is more organic material to break down, consume oxygen, and achieve reducing conditions in the chosen shrimp taxon, which is more than three times heavier than starfish and snails and more than thirty times heavier than a planarian specimen. This emphasises that changes in redox potential are influenced not only by the nature of the decaying organic matter but also by the size of the organism (i.e., the biomass-to-water ratio).

Another nuance that should be emphasised is that these direct comparisons of different organic matter ratios to redox values assume that the decaying organic material of a certain animal is homogenously distributed within the system, which is not the case. For instance, arthropod internal organs, which are rich in proteins, are isolated from the water column by the cuticle, a relatively decay-resistant structure made of complex carbohydrates[4]. This is somewhat similar for more heavily biomineralised animals, such as snails and starfish, since a relatively large portion of the organic material in these organisms is internal to a mineralised skeleton and not readily in contact with water. In other words, during decay, a discrepancy could also be observed between the environment outside the carcass (monitored herein) and the one within the carcass (not monitored herein). In the case of arthropods, for example, the internal environment could be reducing as internal organs decay and consume oxygen, while the external environment remains oxidising as the cuticle has not yet decayed significantly for decaying organic matter to leak outside. In this sense, if arthropod soft tissues were a homogeneous mass, external ORP values would likely have dropped faster than the patterns observed in this experiment (Fig. 1), as proteins in their bodies would have been exposed to water immediately at death. Future experiments should focus on comparing internal versus external variations in redox

conditions. This is particularly relevant since animals and their decay are heterogeneous[21,22,24,26,28,31,33,35,43–54], and data from the fossil record suggest that internal and external soft tissues have different preservation potentials in sites with exceptional fossil preservation[55–60]. Planning such experiment is not straightforward, since measuring redox conditions inside a decaying organism would mean that the body walls have to be broken by the sensor for it to be positioned in the body cavity, or near organs of interest. One possibility to remedy this is in the future would be to compare redox conditions around broken and unbroken samples, which would give a better view of how internal organs influence redox conditions.

## Implication for understanding taphonomic pathways over geological times

A major implication of this study is its insight into the complex patterns of fossil preservation in the rock record. In the current experiment, carcasses of different taxa achieved markedly different redox conditions, even when exposed to similar initial environmental settings. This variation was primarily controlled by the size and composition of the organic material. These results are particularly important because they suggest that different mechanisms of organic matter preservation could operate within a single sedimentary layer. In light of our experimental findings, if a small, protein-rich planarian were to decay in the same sedimentary layer (with limited oxygen supply) as a larger, less protein-rich shrimp, their degradation would proceed under completely different redox conditions even at the same level of burial. For instance, planarian organic material would decay and disappear without significantly affecting oxygen levels (even if oxygen was limited), meaning that oxygen would persist even after the complete degradation of the planarian soft tissues − essentially behaving as if the system were open with an unlimited oxygen supply. In contrast, shrimp organic matter would rapidly consume oxygen during decomposition, leading to further decay under anoxic conditions (behaving as a closed system with limited oxygen). This occurrence of both "closed" and "open" systems within a single sedimentary layer highlights an important but often overlooked process in studies of organic matter degradation and preservation in the rock record. For instance, traditional models of fossil preservation focused on vertical redox zonation within sediments[36]. The uppermost sediment layers, exposed to seawater, are typically oxygen-rich, facilitating processes such as nitrification and organic matter oxidation[61]. At greater depths, denitrification occurs, followed by sulphate reduction, phosphorus release, and methanogenesis[36]. According to these models, the degree of mineralisation a carcass experiences depends on how long it resides within each redox zone, with burial rates controlling the progression from the nitrification to the methanogenesis zone[61]. For instance, prolonged exposure to the sulphate-reducing zone leads to greater pyritisation, while rapid burial may allow a carcass to bypass this zone and enter methanogenesis more quickly[12]. This study suggests that local redox zones in the sediment are more complex and depend on carcass size and organic composition. In theory, two carcasses lying side by side could exhibit entirely different redox conditions in the absence of burial or following burial under identical sediment loads, which essentially means that redox conditions and burial depth are not always linearly related and that lateral variations could occur within the same sedimentary layer. These findings help explain selective authigenic mineralisation patterns in the rock record[14,15,20,62,63], by supporting previously proposed hypotheses suggesting that authigenic mineralisation is taxon- and organ-dependent[62,64,65], with larger carcasses promoting reducing conditions faster than smaller ones[21,62].

A key element to keep in mind is that this experiment began with the presence of oxygen and high ORP values, indicative of oxidative conditions at day 0, which then decreased heterogeneously across different types of animals and organic matter (Fig. 1). The heterogeneity observed among various types of decaying organic matter is

likely to persist under different initial environmental conditions. However, the magnitude of ORP value changes could vary whether different organic matter types were exposed to reducing conditions at day 0. Starting the experiment under reducing conditions means that bacterial communities present at day 0 would differ significantly from those in oxidative conditions. In other words, the results of the current experiment suggest that the size and nature of organic matter play a critical role in controlling oxidation-reduction potential and this observation is unlikely to change with future experiments. However, the magnitude of differences between the different organic matter types and generalisations to natural systems should be made with caution, given the variability in chemical conditions (including but not limited to oxygen) in natural environments[2,7,66–68]. For instance, during the decay of starfish in marine conditions, ORP values were comparable to those observed in freshwater during the first four days of the experiment (Fig. S2). However, from day 5 onward, ORP values around decaying starfish were significantly lower in the presence of saltwater, indicating that organic matter was decomposing and consuming oxygen more rapidly in saltwater than in freshwater (Fig. S2). This may seem unexpected, as salt is generally known to slow down the degradation of organic material in natural systems[69]. However, in the context of this study, this result aligns with the finding that proteins and lipids play a major role in controlling ORP values. In fact, salt affects protein structural integrity, as NaCl increases protein solubility, leading to denaturation and accelerated degradation[70–72]. This degradation results in a more pronounced drop in ORP values.

These findings further emphasise the need for caution when extrapolating the results of this experiment to natural systems, as this was not the intended aim of this study. Additionally, even greater caution should be exercised when considering different geological periods. For example, broadly speaking, the Ediacaran-Cambrian transition, marked by the development of a mixed layer[61] due to pronounced bioturbation[73–75], shifted the positions of redox cycles in the sediment[61]. Processes such as aerobic respiration, which predominantly occurred at the sediment-water interface in the Ediacaran, could have also taken place within sediments in the Cambrian due to penetrative bioturbation, aerating the upper part of the sedimentary column[61]. Other processes, such as calcium phosphate precipitation and sulphate reduction, which were previously confined to the uppermost centimetres of sediment in the Ediacaran, migrated deeper into the sedimentary column and began occurring beneath the mixed layer of the Cambrian sedimentary record[61,73–75]. This example adds to the complexity of interpreting tissue- and taxon-specific redox controls on fossilisation, as any interpretation must account not only for the original chemical (e.g., oxygenation, salinity, etc.) differences between environments but also for the evolution of these chemical conditions over time.

## Methods

The investigation of the redox potential of different animals over time was carried out at the Aquarium Research Lab facility of the Institute of Earth Sciences at the University of Lausanne, Switzerland. Eight freshwater shrimps (*Neocaridina davidi*), twenty freshwater snails (*Stenophysa marmorata*), eight starfish (*Asterina sp.*), and eight planarians (*Procotyla sp.*) were euthanised using one drop of clove oil ($C_7H_{12}ClN_3O_2$) to avoid damaging the specimens. Every organism had reached the adult stage with the shrimps measuring 10–15 mm long and weighing an average of 0.379 g, the snails measuring 10 mm long and weighing an average of 0.0968 g, the starfish measuring 10 mm in diameter and weighing an average of 0.0995 g, and the planarians measuring 10 mm long and weighing 0.0136 g on average. After euthanasia, all the organisms were rinsed thoroughly with reverse osmosis deionised water to ensure that no clove oil remained in the experiment. Forty-four 5 ml crimp top glass vials were then prepared with 3 ml of reverse osmosis deionised water, and each specimen was placed in a separate vial. Starfish were left to decay in freshwater to avoid additional factors that could influence decomposition and redox potential. This approach allowed for the direct comparison of redox potential values across different animals without the confounding effect of salinity. However, eight additional starfish were left to decay separately in saltwater to monitor the effect of salinity on redox potential (see Supplementary Materials, Fig. S2, for details). Eight controls consisted of 3 ml of reverse osmosis water with no animal inside. Each vial was sealed with a plastic lid and placed in the dark at room temperature between 21 °C and 22 °C. The upper part of the vial, between the lid and the water, contained normally oxygenated air from the room. The closure of the vial by a lid minimises the constant flow of new oxidants into the system and is relevant to natural contexts where organic material and carcasses are buried by sediments that restrict oxidant flow from the water column. Such conditions occur in clay-rich deposits, which are often associated with exceptional fossil preservation in the rock record[76,77]. During seven days, the oxidation-reduction potential (ORP in mV) of each sample was measured daily with a microsensor PCE-PHD 1 (PCE Instruments, Germany) using an ORP probe. The microsensor was placed in water ~5 mm above decaying carcasses. In the case of planarians, which decayed and disappeared rapidly in the experiment, the sensor probe was placed ~5 mm above the bottom of the vial.

Since twenty snails were used while for the other animals decaying in freshwater eight specimens were studied, the variance across the raw ORP values in the different animals was assessed with a two-sample t-test to check whether differences in ORP between animals can be explained by the sample size (refer to boxplots in the Supplementary Materials; Fig. S1 & Table S1). The differences in raw ORP values between animals and through time were investigated using a generalised linear model (GLM) and a contrast analysis, where redox potential is the response variable, and animals, time and their interaction are explanatory variables (refer Table S2 in the Supplementary Materials). GLM was chosen as redox potential is a continuous quantitative variable and animals and time (expressed in days) are two qualitative variables[78], and because linear models are robust statistical tests[79]. Because all vials had slightly different ORP values at the start of the experiment, and to remove the impact of these discrepancies in starting values on the interpretation of the results of the experiment at later days, a normalization was done. This has been achieved by dividing all the raw ORP values at the different days by the raw ORP values at time 0 (Fig. 2A). This normalized data, unlike the raw data that is indicative of various chemical conditions and which generally ranges between +350 and −400 mV[6], cannot be used to infer the oxidation-reduction process taking place. The utility of the normalized data is simply to compare whether similar patterns and fluctuations to the raw data are still observed once differences in ORP values between the experimental vials at the start of the experiment have been accounted for. To assess the differences in normalized ORP values between taxa, a second GLM and a contrast analysis were performed with normalized ORP values as the response variable, and animals, time and their interaction as explanatory variables (refer Table S3 in the Supplementary Materials).

To account for the impact of possible contamination in the system, the normalized ORP values of the controls were subtracted from the normalized ORP values of each tested animal. This step isolates the direct impact of the animal organic matter on redox conditions, removing other environmental variables that could influence redox potential – such as the respiration of microbes introduced with the water at day 0 or those that might have contaminated the vial during the experiment preparation. By calculating the difference in normalized ORP values between the control setup (vial and water only) and the experimental setup (animal, vial, and water), a more precise estimate of each animal influence on decay processes is obtained, effectively minimising the effect of confounding variables. Using this data,

we then assessed whether the body mass of the different animals influenced the normalized ORP values, and the redox potentials were corrected based on the animals' weights (Fig. 2B). For example, at a given time point (e.g., day 0), the average normalized ORP value of the controls was subtracted from the normalized ORP value of a shrimp at the same time point, and the result was divided by the average body mass of the shrimp. Another GLM and a contrast analysis were done to assess differences in ORP values through time between the four animal groups while considering the body mass correction, with ORP values as the response variable, and animals, time and their interaction as the explanatory variables (refer Table S4 in the Supplementary Materials). The threshold of significance for all aforementioned analyses was 5%, and all calculated p-values smaller than 0.05 were considered to indicate significantly different redox patterns.

Since the mineralised skeleton of taxa is relatively decay-resistant and unlikely to contribute to changes in ORP values, we repeated the latter analyses by accounting only for the mass of soft tissues in the investigated phyla. Dissecting the organisms to isolate soft tissue would have introduced additional variables related to the structural integrity of the organic material. Therefore, we estimated the mass of soft tissues based on literature data. Usually, when assessing the composition of organisms, taxa are subjected to extreme temperatures, causing organic matter to evaporate. The evaporated components are subsequently isolated, and the proportions of proteins, carbohydrates, and lipids are determined[37–42]. However, mineralised structures, such as skeletal elements, do not evaporate; instead, they transform into ash. By multiplying the proportion of ash in these taxa, based on literature data, by the total mass of the organism, we obtained an estimate of the mass of the skeletal element that is unlikely to contribute to changes in redox values. Then, we subtracted this mass from the total mass of the organisms to obtain a better estimate of the organic material mass, which was then used to plot a new mass-corrected curve (Fig. 2C), organic matter contributing to changes in redox values. To investigate further how organic matter controls ORP values, we represented the estimated content of proteins, lipids, and carbohydrates in each animal as bar plots (Fig. 2D). Additionally, we compared various ratios, including proteins:lipids, carbohydrates:lipids, and (proteins + carbohydrates):lipids, to the average mass-corrected redox value for each taxon to determine how different compounds influence ORP variations (Fig. 2E). Pearson correlation coefficients were calculated for each comparison between ratios and redox values[80] (refer Table S5 in the Supplementary Materials). Note that the proportions of ashes, lipids, carbohydrates, and proteins taken from the literature were based on the taxa studied here. Whenever literature references[37–42] indicated slightly different contents for a certain taxon, we used the average values for our investigations.

All statistical analyses were performed on the software R 4.2.3 (R Core Team, 2021) using the *glm* function for GLM models and *emmeans* package version 1.8.9 for contrast analyses. All the graphs were computed with *ggplot2* package version 3.4.4.

### Reporting summary
Further information on research design is available in the Nature Portfolio Reporting Summary linked to this article.

## Data availability
All data necessary to replicate this work are available in the main text and the supplementary material files. Source data are provided with this paper.

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

## Acknowledgements

Allison Daley is thanked for her comments that helped us improve the manuscript. Sarah Gabbott is also thanked for the fruitful discussions during the initial stages of this work. We thank the Swiss National Science Foundation Ambizione Grant (PZ00P2_209102) for funding NC and FS.

## Author contributions

N.C. and F.S. designed the research. N.C. and F.S. did the experimental work. N.C. performed statistical analyses. N.C. and F.S. interpreted and discussed the data. N.C. made the figures and wrote the initial version of the text with the help of FS and JBA.

## Competing interests

The authors declare no competing interests.
