## [Peer Review file · Nature Communications]

Taxon-specific redox conditions control fossilisation pathways

Corresponding Author: Ms Nora Corthésy

Version 0:

Reviewer comments:

Reviewer #1

(Remarks to the Author)

Please see the attached document.

Reviewer #2

(Remarks to the Author)

In this work, the authors use lab-based experiments to investigate the differences in redox conditions during the decay of a range of taxon in freshwater over a week. They find a link between integral traits of each taxon with variance in local redox potential. The quantification of redox environments allows the authors to identify how each taxa decaying creates conditions that favour different chemical processes, which can be related to preservation and fossilisation pathways. This work demonstrates that taxa can and do generate different localised chemical environments in similar conditions, which impacts the possible preservation regimes available. This provides evidence that fossilisation pathways can vary horizontally within the same site, as well as vertically. These ideas have previously been proposed but have not been tested in such a novel manner.

This is really interesting work that provides quantified evidence for understanding some of the underlying mechanisms of the complex and interlinked processes that control fossilisation. It will help better our knowledge of the drivers of variation within and between fossil sites and the inherent biases in the fossil record. The work is largely well-written and easy to understand, with only some minor errors that can be easily addressed at a later stage. While this work is suitable and deserves to be published, there are some areas I think need addressing beforehand.

The methodology is sound and provides enough detail to be replicated. However, I have a concern over the use of freshwater for the marine starfish (see below), but this could be resolved within the text, rather than requiring further experimental work.

Mainly within the methods, but throughout the work, I get a little lost about which ORP values (raw/normalised/corrected against controls) are being used or referred to. For example, on lines 101-103, I thought the text referred to raw ORP values, but the example (which is very helpful!) used normalised ORP values. It would help the reader to state which ORP value is being referred to at every use, as different values are used in different places.

I think the results and discussion surrounding the composition of the animals need further consideration of the tissues and their interactions with the water. However, this could be easily resolved without the need for further work.

The accessibility of all the figures does need addressing before this manuscript can be published. The figures themselves are nice and support the text well; however, when printed in black and white, the lack of contrast between the colours made it very difficult to read the plots. A simple alteration of changing to uniquely shaped points would help to address this and remove the reliance on colour to differentiate information. I also ask the authors to consider changing the colour scheme to something with more contrast where it is not possible to use different shapes to help with the visual accessibility of the plots.

Overall, this is a well-designed piece of experimental work that has produced results that improve our knowledge of potential

biases in the fossil record and so will be of interest to taphonomists and the field of palaeontology as a whole. I have made some additional minor specific suggestions below for areas where the manuscript could be improved. None of my suggestions require a dramatic amount of change or work, I am very happy to recommend this paper for publication once they have been addressed, especially the concerns over accessibility.

Specific comments:

Supplementary material: I was not able to find the raw data or normalised data for the ORP values in the supplementary material provided. Is this available?

Figure 1 – This figure is lovely, but I encourage you to consider potential accessibility needs. I printed this out in black and white, and it was difficult to distinguish the points. Changing to unique point shapes for each taxon (if not the colours) would really help.

Figure 2 – Similar comment as above for parts a + b. C printed out ok, but a stronger contrast between the colours would help make it easier to read (as would lines between the sections if you prefer).

Supplementary figures – Similar comments, it would be worth reconsidering/checking your colour scheme contrast for usability in grey scale.

Lines 24-26 – Apologies for being very pedantic here but there is an argument to be made that trace fossils also undergo a form of decay, in the sense declining structural integrity, (as well as the other more geological taphonomic processes). Including “bacterial” decay (which is what I think is being referred to) would fix that and make a clearer distinction between the processes that are shared between purely organic fossils (like body/chemical).

Lines 29-33 – I think it would really help to define/give a simple explanation here about what redox conditions are. While I appreciate it is a standard concept, as it's critical to the work that people understand/have knowledge of redox it wouldn't be detrimental to lay it out more explicitly before giving the nice examples. Throughout the work uses both redox potential and redox conditions, and, while it's understandable from context, I think the addition of a definition could prevent any confusion in the reader in what they are and how they are linked.

Similarly, In the methods consider adding a sentence that explains what increase or decrease in ORP or redox potential means in an environmental context. The information is in the text, but a nice, clear statement would help set the reader up to follow the results and interpretation a little more easily.

Line 57 – For me the use of “selected” here implies that not all the animals listed were included in the experiment (pedantic I know!). Consider rewording (“Each organism...”) or just deleting “selected”.

Line 63 – The use of freshwater is sensible for the shrimp/snails/ planarians, but, while I understand why it was used for the starfish to aid comparison, I'm not sure if it was a suitable choice for a marine animal. I think this needs addressing in the text. Do you have a sense of how the change in osmotic potential impacts the decay of the organism and so its ORP values, or if there is any reason to suspect that the starfish would “behave” differently in salt water vs fresh water? I suppose as this work is more concerned with the relative changes in ORP the water wouldn't have a huge impact on the results, however, the unknown difference in conditions between fresh and marine could have some impact on the interpretation of these results in the star fish and their application to real world settings. The comparability of all the results is discussed in the work, but perhaps the disconnect between experimental set up and in-life environment of the starfish should be considered too.

The caveat here is that I am under the impression that these starfish are exclusively marine (perhaps in brackish waters too), so apologies if that is not the case.

Line 65 – Do you have an approximation of the room temperature? It's not an essential addition, it's just always nice to have as it can be subjective between regions.

Line 88 – “Then, to investigate how much changes in ORP values occur daily...” – I think there is a slight readability issue here. I think, from context, it should read something like “Then, to investigate how much change occurs daily in the ORP values...” but please check.

Line 94 – I'm not sure this is excluding the impact of ANY possible contamination, just reducing it (as stated later). Maybe soften the opening here to something like: “To account for the impact of possible contamination...”

Line 111 – Sentence isn't clear. I think changing “to” to “which” will help, e.g. “...because non-mineralised structures are those to degrade rapidly and to control ORP changes shortly after death” -> “because non-mineralised structures are those WHICH degrade rapidly and WHICH control ORP changes shortly after death”

Line 114 – Please add in which packages & functions that were used in R to undertake the stats to improve replicability, results have the potential to vary between packages and versions.

Line 122 – “themselves” – This word can make it a bit hard to follow which taxon is being discussed, I would just delete it.

Line 134 – “... following the values of the controls in which no organic matter was present (Fig. 1).” - I'm totally not sure what

this is trying to convey. I think it means that the planarians ORP values match the values and patterns seen in the controls after day 6, but that is a bit of a guess from context. Please consider re-writing this sentence to make it clearer.

Line 195 – "...without accounting for mineralised parts and moisture.." is this a stand-alone sentence or do all the following percentages of composition not include mineralised parts? The methods highlight that the compositions include non-mineralised parts so I find this sentence is adding ambiguity in to the discussion of composition.

Line 195-207 – I like this approach of using the composition of an animal to understand the drivers behind the chemical environments (and potential environments that could be created). However, when considering the composition of the animals, I think there needs to be an explicit discussion about the "uses" or tissue organisation of the different organic matter. The section almost considers the organisms as homogenous masses of matter. It is clear from the results that composition plays a big part in ORP through chemistry, but there is an interplay with the structural use/tissue types in each taxon and how they decay, which I think needs to be stated/discussed a bit more clearly. The work does broach this with an exploration of the idea that the shrimp are decaying slower because of the lipids, but is the difference in OPR values driven by the total composition or the composition found at the water-animal interface? I.e., if the organisms had been homogenised would the same patterns be recovered due to composition alone, or are things like chitinous shells creating their own localised environments or barriers before they break down to "release" other matter? I think the second option is the underlying take-home message I'm getting from this paragraph, but there is merit in exploring that a bit more and being more explicate.

Line 279 – "This adds..." – The "this" here is a little ambiguous, I'm not sure if it refers to the Ediacaran-Cambrian example or the study as a whole.

Version 1:

Reviewer comments:

Reviewer #1

(Remarks to the Author)

The authors have done a good job responding to both my concerns and to Reviewer #2. I think the manuscript is acceptable in its current form.

Reviewer #2

(Remarks to the Author)

I thank the authors for their considered and detailed responses. The revisions made have improved the strength of the manuscript and fully addressed my comments, including queries about the suitability of a marine organism. The accessibility alternations to the figures are appreciated.

As such, I have no hesitation in recommending this manuscript for publication.

Reviewer #1 (Remarks to the Author):

Reviewer: Julie Bartley, Gustavus Adolphus College

KEY RESULTS—This experimental taphonomy paper examines redox changes in four decaying invertebrates over a period of seven days and concludes that larger, protein-rich organisms reach reducing conditions more rapidly than smaller, lipid-rich organisms. The paper further proposes that these results suggest that inter-taxon differences could lead to different taphonomic pathways in organisms entering the taphonomic window under identical initial conditions.

We would like to thank you for your review and comments, which helped us improve the manuscript.

VALIDITY—The experimental results are intriguing and provide a potentially fruitful set of hypotheses for future work, both in experimental taphonomy and in the fossil record. The predictions are certainly testable. However, the conclusions are not strongly supported by the data available in these experiments. The conclusion states that larger organisms (relative to the water they decompose in) exert a greater influence on redox conditions than smaller organisms (Fig 2A); however, the order of normalized redox impact (greatest to least) is

shrimp>**snail**>**starfish**>planarian

while the mass order (highest mass to lowest) is

shrimp>**starfish**>**snail**>planarian

In addition, the paper concludes that protein content (relative to lipids and carbohydrates) is also an important driver of redox change, such that organisms with higher protein content utilize oxygen faster during decay. The order of protein content (highest to lowest) is (Fig 2C)

Planarian>**starfish**>**snail**>shrimp

while the mass-adjusted redox impact (greatest to least) is

Planarian>**snail**>**starfish**>shrimp

Thus, although the planarian and shrimp follow the proposed pattern, the two intermediate organisms, snails and starfish, do not. Additionally, while the masses of the snail and starfish samples are quite close, their normalized ORP paths are quite different from one another. Based on this reading of the results, it does not appear that the primary conclusions of the paper are well-supported by the data.

We agree with Reviewer 1 that the mass of the organism does not explain everything, especially since a correlation between the mass and organic material is only observed for the two ends of the sequence (shrimp and planarians), as indicated by the Reviewer. For this reason, we tried to look at the chemical compositions of the organisms. And although protein content explains some of the patterns, as pointed out by Reviewer 1, it does not explain everything either. However, in the previous text, and starting in the abstract, we mentioned “larger and protein-rich taxa reaching reducing conditions more rapidly than smaller and lipid-rich ones.” By that, we meant that we also need to look at the lipid content, which correlated with the sequence of mass-adjusted redox impact. However, the plots that we have made in the original manuscript and the choice of words we used made all of this very confusing. In the new version of the paper, we included different plots focusing on organic material ratios for the different organisms (new Figure 2E; see below), and we can see an anti-correlation between the average redox values (corrected for the organic matter mass) for the different animals and their proteins/lipids (Pearson test, $r = -0.9251565$) and (proteins + carbohydrates)/lipids (Pearson test, $r = -0.9990038$) ratios. This negative correlation means that the richer the organism's protein content, the more negative its average redox values will be, corresponding to reducing conditions. This emphasizes that protein-rich taxa reach reducing conditions faster than lipid-rich ones and backs up the conclusions in the original manuscript. Moreover, starting in the abstract and throughout the manuscript, we made changes to the text. For instance, the previously problematic part of the abstract currently reads as follows: “Our results show that different decaying taxa lead to various post-mortem environmental redox conditions. Large carcasses tend to reach reducing conditions more rapidly than smaller ones. However, size does not explain all observed patterns, as environmental redox conditions are also influenced by the nature of the organic material. For instance, taxa with higher proteins-to-lipids and (proteins + carbohydrates)-to-lipids ratios tend to achieve reducing conditions more rapidly than others.” Moreover, the following text was added in the discussion: “When taking the average redox values corrected by organic matter mass (Fig. 2C) and comparing them to different organic material ratios (Fig. 2E), it becomes clear that both proteins and lipids dictate changes in redox conditions more than carbohydrates. For instance, the decrease in ORP values corrected by organic matter mass is accompanied by an increase in proteins-to-lipids (Pearson test, $r = -0.9251565$, Tab. S5) and (proteins + carbohydrates)-to-lipids (Pearson test, $r = -0.9990038$, Fig. 2F, Tab. S5) ratios (Fig. 2E). This essentially means that protein quantities are anti-correlated to mass-corrected ORP values (Fig. 2F), and that decreasing ORP values and achieving reducing conditions is easier with higher protein quantities and lower lipid values. On the contrary, when accounting for carbohydrates on their own (e.g., carbohydrates-to-lipids ratio), no correlation is observed with average redox values corrected by organic matter mass (Fig. 2E, $r = -0.2538133$, Fig. 2F, Tab. S5), indicating that the role of carbohydrates in influencing redox conditions is ambiguous. These results indicate that organisms with a higher proportion of lipids and a lower proportion of proteins break down and consume oxygen at slower rates than those with a low proportion of lipids and a high proportion of proteins, leading to differences in ORP values (Fig. 2E). This may explain why, when correcting for size differences, shrimps show the least dropdown in ORP values, followed by starfish, snails, and planarians (Fig. 2C, E).”

That said, the results are suggestive. Specifically, when the carcass mass:water ratio is high, it is easier for redox potential to change quickly. However, this is a matter of simple stoichiometry – oxygen becomes a limiting reactant under these conditions. Natural systems are not closed in this way, though they may be diffusion-limited and thus act somewhat like a closed system. It would not be unreasonable to propose that a larger carcass might experience diffusion-limited local redox change more quickly than a smaller carcass. It's possible, perhaps, that the authors could probe that relationship by examining the individual carcasses by mass to ascertain whether they show a pattern related to size. Such an analysis would likely have limited statistical support, but a suggestive pattern might be interpretable in light of the overall findings.

We agree with Reviewer 1 that the observed data primarily reflect stoichiometric processes and that our experimental design does not replicate natural systems — nor is this its aim, as such replication is simply not possible in a laboratory setting. However, as the Reviewer pointed out, some natural systems can function as closed systems, particularly in sites where soft tissue preservation occurs. Fossils in these sites are often buried in clay-rich sediments (see, for instance, Anderson et al., 2018; Saleh et al., 2019) that limit the continuous supply of oxygen to the carcass. While our experimental design does not replicate a natural setting, it remains relevant in the context of soft tissue preservation, which is the primary focus of our discussion. That said, we acknowledge that our previous wording — particularly at the beginning of the section *Implications for Understanding Taphonomic Pathways Over Geological Time* — was problematic. It inadvertently implied that burial is not required in natural settings to achieve reducing conditions, as if a small decaying carcass could reach reducing conditions while fully

exposed to the open water column. This was inaccurate, and we have removed the misleading text from the revised manuscript. Regarding the mass analyses suggested by the Reviewer, we do not believe they are necessary. As stated in response to the previous comment, we agree with Reviewer 1 that mass alone does not explain everything, and we have removed ambiguous text that previously suggested otherwise. To improve precision, we have added the following text to the manuscript: “The closure of the vial by a lid minimizes the constant flow of new oxidants into the system and is relevant to natural contexts where organic material and carcasses are buried by sediments that restrict oxidant flow from the water column. Such conditions occur in clay-rich deposits, which are often associated with exceptional fossil preservation in the rock record (Anderson et al., 2018; Saleh et al., 2019).”

SIGNIFICANCE—The authors correctly point out that taphonomic variations, even among organisms buried under similar conditions (e.g., the same bedding plane), are very difficult to explain. This approach and these preliminary results represent a useful way to pursue this question. I think it is likely that the conclusion of this paper is basically correct; my concern is that the data do not demand this explanation and I do not think the uncertainty is fully addressed in the article.

As indicated in our responses to previous comments, as well as in responses to later comments in this review and to those made by Reviewer 2, and after incorporating new data and analyses while removing ambiguous text, we are confident that all our conclusions in the revised manuscript are well supported by the data.

DATA AND METHODOLOGY—The methodology suffers from a crucial flaw, and that is that the experimental systems are “closed” to varying degrees. The average shrimp carcass is more than 25x larger than the average planarian, and both were decomposed in the same volume of water (and thus the same effective O₂ concentration). It is likely that the planarian chamber remained an open system with respect to the carcass throughout the experiment, while the shrimp carcass exhibited closed-system behavior for nearly the entire experiment duration. Unfortunately, the only remedy I see for this flaw is to repeat the experiment with either similar biomass in each chamber or oxygen availability scaled to carcass size (i.e., use more water for larger carcass sizes). When reviewing manuscripts, I try very hard not to send the authors back to the bench, but here I’m not seeing much of an alternative. My suggestion above (examining individual carcasses by mass) might mitigate the issue, but likely would not solve it.

This is an interesting point raised by Reviewer 1. However, what the Reviewer sees as a major limitation of the paper is actually one of its most important implications. Here, we do not have systems that are “closed” to varying degrees, as the experimental chamber and design remain constant — the only variable in the system is the type of animal used. To properly test whether different animals influence redox conditions differently, the chamber size must remain the same; otherwise, introducing a new variable would make the experiment inconsistent and difficult to interpret. Ideally, we would compare different taxa of similar dimensions within the same chamber size. However, this is currently not feasible, as we do not have organisms of the same size across taxa — particularly planarians, which are minuscule compared to the other organisms, especially shrimps. Where we do agree with Reviewer 1 is in recognizing that these differences in size influenced the experimental outcomes. Specifically, the limited availability of organic material in the planarian led to rapid decay without significantly affecting oxygen levels, as the ratio of water to organic matter was high. This suggests that the planarian decayed in a system with abundant oxygen (an open system). In contrast, the much larger shrimp carcass consumed oxygen rapidly, and in the absence of new oxygen input, the shrimp experiment effectively functioned as a closed system. Rather than being a limitation, this is a key observation that we only briefly hinted at in the previous version of the manuscript but did not fully discuss. Accounting for differences in the water-to-organic matter ratio implies that if a planarian and a shrimp were decaying in the same sedimentary layer — composed of sediments that allow only limited oxygen supply — the planarian would decay in the presence of oxygen, which would remain in the surrounding environment even after the organic material had completely disappeared. In contrast, the shrimp would rapidly consume oxygen and continue to decay under reducing conditions. These disparities between planarians and arthropods would lead not only to different probabilities of preservation but also to distinct taphonomic pathways. For instance, cases of soft tissue mineralization, such as pyritization, would only be expected in the arthropod scenario. To address this, we have included the following text in the discussion: “A major implication of this study is its insight into the complex patterns of fossil preservation in the rock record. In the current experiment, carcasses of different taxa achieved markedly different redox conditions, even when exposed to similar initial environmental settings. This variation was primarily controlled by the size and composition of the organic material. These results are particularly important because they suggest that different mechanisms of organic matter preservation could operate within a single sedimentary layer. In light of our experimental findings, if a small, protein-rich planarian were to decay in the same sedimentary layer (with limited oxygen supply) as a larger, less protein-rich shrimp, their degradation would proceed under completely different redox conditions even at the same level of burial. For

instance, the planarian's organic material would decay and disappear without significantly affecting oxygen levels (even if oxygen was limited), meaning that oxygen would persist even after the complete degradation of the planarian's soft tissues — essentially behaving as if the system were open with an unlimited oxygen supply. In contrast, the shrimp's organic matter would rapidly consume oxygen during decomposition, leading to further decay under anoxic conditions (behaving as a closed system with limited oxygen). This occurrence of both “closed” and “open” systems within a single sedimentary layer highlights an important but often overlooked process in studies of organic matter degradation and preservation in the rock record. [...]

I have a few smaller questions about methodology that are probably easily addressed by the authors:

Does the reported mass include skeletons? If not, please specify that and specify how you determined skeletal mass separately from soft tissue mass. If the skeleton is included in the reported mass, it would be even harder for me to support the conclusion of the mass-redox relationship, as the skeletons aren't likely participating in the redox reactions.

In the original manuscript, the mass included skeletal elements because dissecting the animals to isolate their skeletons without damaging the specimens was challenging due to their small sizes. However, since we agree with Reviewer 1 that the skeleton is unlikely to participate in redox reactions, we have adopted a new approach that accounts only for the organic matter mass. Using literature data, we obtained information on the organic versus mineralogical content of these taxa and created a new plot (Figure 2C) to reflect this adjustment. While we observe a slight shift in some mass-adjusted redox curves, the order of redox changes between taxa remains unchanged (see figure below). We have also included the following text in the *Materials and Methods* section: “Since the mineralized skeleton of taxa is relatively decay-resistant and unlikely to contribute to changes in ORP values, we repeated the latter analyses by accounting only for the mass of soft tissues in the investigated phyla. Dissecting the organisms to isolate soft tissues would have introduced additional variables related to the structural integrity of the organic material. Therefore, we estimated the mass of soft tissues based on literature data. Usually, when assessing the composition of organisms, taxa are subjected to extreme temperatures, causing organic matter to evaporate. The evaporated components are subsequently isolated, and the proportions of proteins, carbohydrates, and lipids are determined (Barrett & Butterworth, 1982; Debnath et al., 2016; Doxa et al., 2013; Jung, 2002; Namaei Kohal et al., 2018; Tomas et al., 2020). However, mineralized structures, such as skeletal elements, do not evaporate; instead, they transform into ash. By multiplying the proportion of ash in these taxa, based on literature data, by the total mass of the organism, we obtained an estimate of the mass of the skeletal element that is unlikely to contribute to changes in redox values. Then, we subtracted this mass from the total mass of the organisms to obtain a better estimate of the organic material mass, which was then used to plot a new mass-corrected curve (Fig. 2C), organic matter contributing to changes in redox values. To investigate further how organic matter controls ORP values, we represented the estimated content of proteins, lipids, and carbohydrates in each animal as bar plots (Fig. 2D). Additionally, we compared various ratios, including proteins:lipids, carbohydrates:lipids, and (proteins + carbohydrates):lipids, to the average mass-corrected redox value for each taxon to determine how different compounds influence ORP variations (Fig. 2E). Pearson correlation coefficients were calculated for each comparison between ratios and redox values (Benesty et al., 2009) (refer Tab. S4 in the *Supplementary Materials*). Note that the proportions of ashes, lipids, carbohydrates, and proteins taken from the literature were based on the taxa studied here. Whenever literature references (Barrett & Butterworth, 1982; Debnath et al., 2016; Doxa et al., 2013; Jung, 2002; Namaei Kohal et al., 2018; Tomas et al., 2020) indicated slightly different contents for a certain taxon, we used the average values for our investigations.”

Line 111 indicates that proportions of proteins, lipids, and carbohydrates were obtained from the literature. Were the taxa used in the experiment the same taxa analyzed in the literature sources? I don't think it's likely to matter very much, but this should be clarified.

Yes, they were the same taxa. Whenever different studies pointed to slightly different compositions, we calculated the average of these compositions and used it in the manuscript. This has been now clearly stated in the material and methods section: "Note that the proportions of ashes, lipids, carbohydrates, and proteins taken from the literature were based on the taxa studied here. Whenever literature references indicated slightly different contents for a certain taxon, we used the average values for our investigations."

Was the head space in each vial filled with fully oxygenated air? It is worth specifying that condition as well.

Yes, this was the case and this is now clearly stated in the material and methods section: "The upper part of the vial, between the lid and the water, contained fully oxygenated air from the room."

ANALYTICAL APPROACH—Statistics is not my area of expertise, but there are a few small items that would clarify the analyses:

Figure S1 should specify the variance across what variable? I am thinking it is ORP, but please be explicit.

Yes, the variance corresponds to the variance across the raw ORP values. We added this information in the caption of Figure S1.

Specify the threshold for significance. It looks like differences are significant at $p < 0.05$, but that should be stated.

Yes, this was the case, and this is now indicated in the material and methods section: “The threshold of significance for all aforementioned analyses was 5%, and all calculated p-values smaller than 0.05 were considered to indicate significantly different redox patterns.”

I don’t understand the utility of the analysis of daily change in ORP values. First of all, reversing the sign on the “normalized ORP daily difference variable” such that this variable is positive when the observed ORP value goes *down* is unnecessary and confusing (I spent much too long trying to figure that out, even though it is stated in the figure caption). Sign of the variable aside, this analysis does not seem to contribute anything to the overall conclusions.

We do agree, and this figure was removed from the *Supplementary Materials*, and its corresponding text from the *Material and Methods* section.

In Figure 1 and figures 2A, 2B, the figure caption needs to explain what the line represents. It appears to be smoothed, rather than a mean line. And presumably, the dots are the individual measurements, but it would be good to be explicit about that as well.

The captions of Figures 1 and 2 were edited to explain what the points and lines represent on the graphs. “The shaped points represent individual measurements of redox potential and the lines are following the average redox potential of each animal at each time point.”

SUGGESTED IMPROVEMENTS—I think I’ve summarized possible improvements to the manuscript in the categories above. As I said, I really hesitate to tell authors to perform additional experiments, so perhaps analyzing the mass-ORP relationships a little more closely would provide additional support for the hypothesis.

We would like to thank Reviewer 1 for her constructive criticism, which helped us improve the manuscript. We are confident that we addressed all her comments by adding new analyses and considerably improving the text.

CLARITY AND CONTEXT—Overall, the manuscript is clearly written. I have a few suggestions organized by line number:

L69 Change *sensor* to **sensor**

Done

L70 the variance **in ORP** [if indeed it is the variance in ORP that’s being analyzed]

Done

L105 Does the body mass include the skeleton?

This has been addressed, as explained on the previous page.

L125 Change sample size to **number of samples** [to avoid confusion with body mass]

Done

L163 change In this sense to **in this situation**

Done

L166 change frees to **releases**

Done

L188-191 I don't think you can draw that conclusion quite so strongly, because of the closed-system behavior likely experienced by the shrimp. Also, snails have less protein than starfish, so why would they experience a more marked drop in ORP, according to the proposed explanation?

This has been now addressed in the new version of the manuscript as indicated in the response to previous comments. We emphasized throughout the new version of the manuscript the fact that size does not explain everything, the difference between open and closed systems, and the fact that one should look not only at proteins but at lipids as well. This sentence was also purely theoretical, and it was written before presenting the chemical composition data (on proteins, lipids, etc.) we made this even clearer in the new version of the text.

L192 change proximate to **approximate**

Done

L192 remove , near the end of this line

Done

Fig 2 change proximate on the y-axis to **approximate**

Done

L229-230 This is only true in a closed nutrient system. On the seafloor, we would expect water to be circulating over the tops of these carcasses, so in this sense, the experiment is not a good model for pre-burial taphonomy; it is a better model for early post-burial, where diffusion limits the supply of O₂ to the carcass.

We agree, and we removed this part from the discussion. We accounted for this comment throughout the manuscript, as indicated in the response to previous comments in this document.

L262 change organic matters to **organic matter types**

Done

References

In several references, genus and species names are not in italics. See, for example, ref #s 6, 15, 18, 35, 39, 45.

Ref #26: *pap* needs to be capitalized

Most references use the full journal title, but a few use an abbreviation (cf. 10, 22, 26, 35). Probably, they should all be the same.

Done

Reviewer #2 (Remarks to the Author):

In this work, the authors use lab-based experiments to investigate the differences in redox conditions during the decay of a range of taxa in freshwater over a week. They find a link between integral traits of each taxon with variance in local redox potential. The quantification of redox environments allows the authors to identify how each taxon decaying creates conditions that favour different chemical processes, which can be related to preservation and fossilisation pathways. This work demonstrates that taxa can and do generate different localised

chemical environments in similar conditions, which impacts the possible preservation regimes available. This provides evidence that fossilisation pathways can vary horizontally within the same site, as well as vertically. These ideas have previously been proposed but have not been tested in such a novel manner. This is really interesting work that provides quantified evidence for understanding some of the underlying mechanisms of the complex and interlinked processes that control fossilisation. It will help better our knowledge of the drivers of variation within and between fossil sites and the inherent biases in the fossil record. The work is largely well-written and easy to understand, with only some minor errors that can be easily addressed at a later stage. While this work is suitable and deserves to be published, there are some areas I think need addressing beforehand.

We would like to thank Reviewer 2 for the constructive review and comments that helped us improve the manuscript.

The methodology is sound and provides enough detail to be replicated. However, I have a concern over the use of freshwater for the marine starfish (see below), but this could be resolved within the text, rather than requiring further experimental work.

We acknowledge Reviewer 2's concern regarding the decomposition of starfish in freshwater, given that they are marine organisms. In our study, we focused on redox potential changes during the decomposition of different organic matter in freshwater only, as we aimed to avoid introducing additional environmental factors such as water salinity. Additionally, decomposing starfish in freshwater ensured comparability with other animals in our study, as salinity can significantly influence decomposition (Corthésy et al., 2025; Fraga & Vega, 2025; Gäb et al., 2020). To address Reviewer 2's comments, we conducted new experiments in which starfish were decomposed in saltwater to assess its impact on redox potential. We found that at the start of the experiment, ORP values were comparable between freshwater and marine conditions. However, from day 4 onward, ORP values were consistently lower in saltwater than in freshwater (see figure below). This can be attributed to the role of sodium and chloride, key constituents of salt, in accelerating protein degradation by affecting their structural integrity (Gonzalez & Bradley, 1995). For example, NaCl increases protein solubility, leading to their denaturation and faster breakdown (Suliman et al., 2006; Trevino et al., 2008). In this context, the accelerated decomposition of proteins in marine salinity likely explains the progressively lower ORP values in saltwater compared to freshwater. We have now included this data in a new figure in the *Supplementary Materials* (new Fig. S2). However, we chose not to incorporate it into the main text, as it would introduce a new variable and create inconsistencies as we do not have equivalent saltwater decomposition data for the other animals in our study, all of which (including shrimps, planarians, and snails) were freshwater organisms. To clarify this limitation, we added the following statement in the *Material and Methods* section: "Starfish were left to decay in freshwater to avoid additional factors that could influence decomposition and redox potential. This approach allowed for the direct comparison of redox potential values across different animals without the confounding effect of salinity. However, eight additional starfish were left to decay separately in saltwater to monitor the effect of salinity on redox potential (see *Supplementary Materials*, Fig. S2, for details)." Additionally, we added the following clarification to the *Discussion* section: "However, the magnitude of differences between the different organic matter types and generalisations to natural systems should be made with caution, given the variability in chemical conditions (including but not limited to oxygen) in natural environments (Allison & Briggs, 1993; Gaines, 2014; Muscente et al., 2017; Saleh, Vaucher, et al., 2021; Saleh, Qi, et al., 2022). For instance, during the decay of starfish in marine conditions, ORP values were comparable to those observed in freshwater during the first four days of the experiment (Fig. S2). However, from day 5 onward, ORP values around decaying starfish were significantly lower in the presence of saltwater, indicating that organic matter was decomposing and consuming oxygen more rapidly in saltwater than in freshwater (Fig. S2). This may seem unexpected, as salt is generally known to slow down the degradation of organic material in natural systems (Wijnker et al., 2006). However, in the context of this study, this result aligns with the finding that proteins and lipids play a major role in controlling ORP values. In fact, salt affects protein structural integrity, as NaCl increases protein solubility, leading to denaturation and accelerated degradation (Gonzalez & Bradley, 1995; Suliman et al., 2006; Trevino et al., 2008). This degradation results in a more pronounced drop in ORP values."

Figure S2. Oxidation-reduction potential (ORP) values (mV) over time for starfish decaying in freshwater and saltwater. ORP values reflect different redox zonations: nitrification (+100 to +350 mV), carbonaceous biochemical oxygen demand (cBOD) (+50 to +250 mV), denitrification (-50 to +50 mV), sulphate reduction (-250 to -50 mV), biological phosphorus release (-250 to -100 mV), and methanogenesis (-400 to -175 mV).

Mainly within the methods, but throughout the work, I get a little lost about which ORP values (raw/normalised/corrected against controls) are being used or referred to. For example, on lines 101-103, I thought the text referred to raw ORP values, but the example (which is very helpful!) used normalised ORP values. It would help the reader to state which ORP value is being referred to at every use, as different values are used in different places.

We have taken this comment into account and made changes to the text to make it clear which ORP values we are referring to each time.

I think the results and discussion surrounding the composition of the animals need further consideration of the tissues and their interactions with the water. However, this could be easily resolved without the need for further work.

We have added an entire section on this particular issue, as indicated on the last page of this document.

The accessibility of all the figures does need addressing before this manuscript can be published. The figures themselves are nice and support the text well; however, when printed in black and white, the lack of contrast between the colours made it very difficult to read the plots. A simple alteration of changing to uniquely shaped points would help to address this and remove the reliance on colour to differentiate information. I also ask the authors to consider changing the colour scheme to something with more contrast where it is not possible to use different shapes to help with the visual accessibility of the plots.

We agree with Reviewer 2 about this issue with the figures. We have edited all the figures by adding shaped points so that they are easier to read.

Overall, this is a well-designed piece of experimental work that has produced results that improve our knowledge of potential biases in the fossil record and so will be of interest to taphonomists and the field of palaeontology as

a whole. I have made some additional minor specific suggestions below for areas where the manuscript could be improved. None of my suggestions require a dramatic amount of change or work, I am very happy to recommend this paper for publication once they have been addressed, especially the concerns over accessibility. We would like to thank Reviewer 2 for the constructive and encouraging comments. We have accounted for their suggestions and edited the manuscript accordingly.

Specific comments:

Supplementary material: I was not able to find the raw data or normalised data for the ORP values in the supplementary material provided. Is this available?

This is an omission on our behalf. The raw data are now added in a separate Excel sheet.

Figure 1 – This figure is lovely, but I encourage you to consider potential accessibility needs. I printed this out in black and white, and it was difficult to distinguish the points. Changing to unique point shapes for each taxon (if not the colours) would really help.

Figure 1 is now improved to ensure accessibility.

Figure 2 – Similar comment as above for parts a + b. C printed out ok, but a stronger contrast between the colours would help make it easier to read (as would lines between the sections if you prefer). Figure 2 is now improved to ensure accessibility.

Supplementary figures – Similar comments, it would be worth reconsidering/checking your colour scheme contrast for usability in grey scale.

Supplementary figures are now improved to ensure accessibility.

Lines 24-26 – Apologies for being very pedantic here but there is an argument to be made that trace fossils also undergo a form of decay, in the sense declining structural integrity, (as well as the other more geological taphonomic processes). Including “bacterial” decay (which is what I think is being referred to) would fix that and make a clearer distinction between the processes that are shared between purely organic fossils (like body/chemical).

Done

Lines 29-33 – I think it would really help to define/give a simple explanation here about what redox conditions are. While I appreciate it is a standard concept, as it’s critical to the work that people understand/have knowledge of redox it wouldn’t be detrimental to lay it out more explicitly before giving the nice examples. Throughout the work uses both redox potential and redox conditions, and, while it’s understandable from context, I think the addition of a definition could prevent any confusion in the reader in what they are and how they are linked.

We have added text for more context and to differentiate between redox potential and redox condition: “Redox condition refers to the overall oxidation-reduction state of an environment, which is determined by the balance between oxidizing and reducing processes. It is influenced by the presence and activity of electron donors (reducing agents) and electron acceptors (oxidizing agents). The redox condition of an environment can be deciphered by investigating the redox potential or oxidation-reduction potential (ORP) of this environment. The redox potential is a measure of the tendency of a chemical species to gain or lose electrons in a redox reaction (Reichert et al., 2007). It is typically expressed in millivolts (mV) and indicates whether a system is more oxidizing (positive ORP) or reducing (negative ORP)(Al-Hussainin, 2016).”

Similarly, In the methods consider adding a sentence that explains what increase or decrease in ORP or redox potential means in an environmental context. The information is in the text, but a nice, clear statement would help set the reader up to follow the results and interpretation a little more easily.

We chose to include this information in the introduction of the paper (see reply to the comment above), as if the paper is accepted, the *Material and Methods* section will be at the end of the manuscript. Therefore, the only way to ensure this information is readily accessible to readers is by adding it to the introduction.

Line 57 – For me the use of “selected” here implies that not all the animals listed were included in the experiment (pedantic I know!). Consider rewording (“Each organism...”) or just deleting “selected”.

The word “selected” was removed from the sentence.

Line 63 – The use of freshwater is sensible for the shrimp/snails/ planarians, but, while I understand why it was used for the starfish to aid comparison, I’m not sure if it was a suitable choice for a marine animal. I think this needs addressing in the text. Do you have a sense of how the change in osmotic potential impacts the decay of the organism and so its ORP values, or if there is any reason to suspect that the starfish would “behave” differently in salt water vs fresh water? I suppose as this work is more concerned with the relative changes in ORP the water wouldn’t have a huge impact on the results, however, the unknown difference in conditions between fresh and marine could have some impact on the interpretation of these results in the star fish and their application to real world settings. The comparability of all the results is discussed in the work, but perhaps the disconnect between experimental set up and in-life environment of the starfish should be considered too. The caveat here is that I am under the impression that these starfish are exclusively marine (perhaps in brackish waters too), so apologies if that is not the case.

This is indeed an important issue. To address Reviewer 2’s concerns, we have added new data in the *Supplementary Materials*, along with additional text in the *Materials and Methods* and *Discussion* sections, as detailed in our response to a previous comment by the reviewer.

Line 65 – Do you have an approximation of the room temperature? It’s not an essential addition, it’s just always nice to have as it can be subjective between regions.

The room temperature was maintained between 21°C and 22°C. We have added this detail to the *Materials and Methods* section.

Line 88 – “Then, to investigate how much changes in ORP values occur daily...” – I think there is a slight readability issue here. I think, from context, it should read something like “Then, to investigate how much change occurs daily in the ORP values...” but please check.

This data analysis was removed following comments from Reviewer 1, so this is no longer an issue.

Line 94 – I’m not sure this is excluding the impact of ANY possible contamination, just reducing it (as stated later). Maybe soften the opening here to something like: “To account for the impact of possible contamination...” We have taken this suggestion into account.

Line 111 – Sentence isn’t clear. I think changing “to” to “which” will help, e.g. “...because non-mineralised structures are those to degrade rapidly and to control ORP changes shortly after death” -> “because non-mineralised structures are those WHICH degrade rapidly and WHICH control ORP changes shortly after death”

This entire section was reworded to account for other comments raised by Reviewer 1.

Line 114 – Please add in which packages & functions that were used in R to undertake the stats to improve replicability, results have the potential to vary between packages and versions. R packages and functions were added to the methods section: “All statistical analyses were performed on the software R 4.2.3 (R Core Team, 2021) using the *glm* function for GLM models and *emmeans* package version 1.8.9 for contrast analyses. All the graphs were computed with *ggplot2* package version 3.4.4.”

Line 122 – “themselves” – This word can make it a bit hard to follow which taxon is being discussed, I would just delete it.

Done

Line 134 – “... following the values of the controls in which no organic matter was present (Fig. 1).” - I’m totally not sure what this is trying to convey. I think it means that the planarians ORP values match the values and patterns

seen in the controls after day 6, but that is a bit of a guess from context. Please consider re-writing this sentence to make it clearer.

This sentence was reworded.

Line 195 – "...without accounting for mineralised parts and moisture.." is this a stand-alone sentence or do all the following percentages of composition not include mineralised parts? The methods highlight that the compositions include non-mineralised parts so I find this sentence is adding ambiguity in to the discussion of composition.

This part of the sentence was removed, and further details on what was used in these comparisons were added to the *Material and Methods* section, as explained in our response to Reviewer 1.

Line 195-207 – I like this approach of using the composition of an animal to understand the drivers behind the chemical environments (and potential environments that could be created). However, when considering the composition of the animals, I think there needs to be an explicit discussion about the "uses" or tissue organisation of the different organic matter. The section almost considers the organisms as homogenous masses of matter. It is clear from the results that composition plays a big part in ORP through chemistry, but there is an interplay with the structural use/tissue types in each taxon and how they decay, which I think needs to be stated/discussed a bit more clearly. The work does broach this with an exploration of the idea that the shrimp are decaying slower because of the lipids, but is the difference in OPR values driven by the total composition or the composition found at the water-animal interface? I.e., if the organisms had been homogenised would the same patterns be recovered due to composition alone, or are things like chitinous shells creating their own localised environments or barriers before they break down to "release" other matter? I think the second option is the underlying take-home message I'm getting from this paragraph, but there is merit in exploring that a bit more and being more explicate.

Indeed, this is a great point raised by the reviewer and we have added the following text in the discussion to account for these elements: "Another nuance that should be emphasized is that these direct comparisons of different organic matter ratios to redox values assume that the decaying organic material of a certain animal is homogeneously distributed within the system, which is not the case. For instance, arthropod internal organs, which are rich in proteins, are isolated from the water column by the cuticle, a relatively decay-resistant structure made of complex carbohydrates (Briggs, 2003). This is somewhat similar for more heavily biomineralized animals, such as snails and starfish, since a relatively large portion of the organic material in these organisms is internal to a mineralised skeleton and not readily in contact with seawater. In other words, during decay, a discrepancy could also be observed between the environment outside the carcass (monitored herein) and the one within the carcass (not monitored herein). In the case of arthropods, for example, the internal environment could be reducing as internal organs decay and consume oxygen, while the external environment remains oxidizing as the cuticle has not yet decayed significantly for decaying organic matter to leak outside. In this sense, if arthropod soft tissues were a homogeneous mass, external ORP values would likely have dropped faster than the patterns observed in this experiment (Fig. 1), as proteins in their bodies would have been exposed to water immediately at death. Future experiments should focus on comparing internal versus external variations in redox conditions. This is particularly relevant since animals and their decay are heterogeneous (Allison, 1986, 1988; Bath Enright et al., 2021; Briggs & Kear, 1993; Butler et al., 2015; Clements et al., 2022; Corthésy et al., 2025; Gäb et al., 2020; Hof & Briggs, 1997; Iniesto et al., 2013, 2015, 2016, 2017; Murdock et al., 2014; Naimark et al., 2016; Sansom, 2016; Sansom et al., 2010b, 2010a, 2013; Wilson et al., 2014), and data from the fossil record suggest that internal and external soft tissues have different preservation potentials in sites with exceptional fossil preservation (Saleh, Bath-Enright, et al., 2021; Saleh et al., 2020, 2023, 2024; Saleh, Ma, et al., 2022; Whitaker et al., 2022). Planning such experiment is not straightforward, since measuring redox conditions inside a decaying organism would mean that the body walls have to be broken by the sensor for it to be positioned in the body cavity, or near organs of interest. One possibility to remedy this in the future would be to compare redox conditions around broken and unbroken samples, which would give a better view on how internal organs influence redox conditions."

Line 279 – "This adds..." – The "this" here is a little ambiguous, I'm not sure if it refers to the Ediacaran-Cambrian example or the study as a whole.

We meant the Ediacaran-Cambrian transition example. We made this clearer in the new version of the text.

REFERENCES

Al-Hussainin, S. N. H. (2016). The oxidation reduction potential distribution along Diyala river within Baghdad city. *Mesopotamia Environmental Journal*, 2, 54-66.

Allison, P. A. (1986). Soft-bodied animals in the fossil record : The role of decay in fragmentation during transport. *Geology*, 14(12), 979. [https://doi.org/10.1130/0091-7613\(1986\)14<979:SAITFR>2.0.CO;2](https://doi.org/10.1130/0091-7613(1986)14<979:SAITFR>2.0.CO;2)

Allison, P. A. (1988). The role of anoxia in the decay and mineralization of proteinaceous macro-fossils. *Paleobiology*, 14(2), 139-154. <https://doi.org/10.1017/S009483730001188X>

Allison, P. A., & Briggs, D. E. G. (1993). Exceptional fossil record : Distribution of soft-tissue preservation through the Phanerozoic. *Geology*, 21(6), 527-530. [https://doi.org/10.1130/0091-7613\(1993\)021<0527:EFRDOS>2.3.CO;2](https://doi.org/10.1130/0091-7613(1993)021<0527:EFRDOS>2.3.CO;2)

Anderson, R. P., Tosca, N. J., Gaines, R. R., Mongiardino Koch, N., & Briggs, D. E. G. (2018). A mineralogical signature for Burgess Shale-type fossilization. *Geology*, 46(4), 347-350. <https://doi.org/10.1130/G39941.1>

Barrett, J., & Butterworth, P. E. (1982). Carbohydrate and lipid catabolism in the planarian *Polycelis nigra*. *Journal of Comparative Physiology*, 146(1), 107-112. <https://doi.org/10.1007/BF00688723>

Bath Enright, O. G., Minter, N. J., Sumner, E. J., Mángano, M. G., & Buatois, L. A. (2021). Flume experiments reveal flows in the Burgess Shale can sample and transport organisms across substantial distances. *Communications Earth & Environment*, 2(1), 1-7. <https://doi.org/10.1038/s43247-021-00176-w>

Benesty, J., Chen, J., Huang, Y., & Cohen, I. (2009). Pearson Correlation Coefficient. In I. Cohen, Y. Huang, J. Chen, & J. Benesty (Eds.), *Noise Reduction in Speech Processing* (p. 1-4). Springer. https://doi.org/10.1007/978-3-642-00296-0_5

Briggs, D. E. G. (2003). The Role of Decay and Mineralization in the Preservation of Soft-Bodied Fossils. *Annual Review of Earth and Planetary Sciences*, 31(1), 275-301. <https://doi.org/10.1146/annurev.earth.31.100901.144746>

Briggs, D. E. G., & Kear, A. J. (1993). Decay and preservation of polychaetes : Taphonomic thresholds in soft-bodied organisms. *Paleobiology*, 19(1), 107-135. <https://doi.org/10.1017/S0094837300012343>

Butler, A. D., Cunningham, J. A., Budd, G. E., & Donoghue, P. C. J. (2015). Experimental taphonomy of *Artemia* reveals the role of endogenous microbes in mediating decay and fossilization. *Proceedings of the Royal Society B: Biological Sciences*, 282(1808), 20150476. <https://doi.org/10.1098/rspb.2015.0476>

Clements, T., Purnell, M. A., & Gabbott, S. (2022). Experimental analysis of organ decay and pH gradients within a carcass and the implications for phosphatization of soft tissues. *Palaeontology*, 65(4), e12617. <https://doi.org/10.1111/pala.12617>

Corthésy, N., Saleh, F., Antcliff, J. B., & Daley, A. C. (2025). Kaolinite induces rapid authigenic mineralisation in unburied shrimps. *Communications Earth & Environment*, 6(1), 1-8. <https://doi.org/10.1038/s43247-024-01983-7>

Debnath, C., Sahoo, L., Haldar, A., Datta, M., Yadav, G., Singha, A., & Bhattacharjee, J. (2016). Proximate and Mineral Composition of Freshwater Snails of Tripura, North-East India. *Fishery Technology*, 53, 307-312.

Doxa, C. K., Divanach, P., & Kentouri, M. (2013). Consumption rates and digestibility of four food items by the marine gastropod *Charonia sequeenzae* (Aradas & Benoît, 1870). *Journal of Experimental Marine Biology and Ecology*, 446, 10-16. <https://doi.org/10.1016/j.jembe.2013.04.019>

Fraga, M. C., & Vega, C. S. (2025). Decay and preservation in marine basins : A guide to small multi-element skeletons. *International Biodeterioration & Biodegradation*, 196, 105904. <https://doi.org/10.1016/j.ibiod.2024.105904>

Gäb, F., Ballhaus, C., Stinnesbeck, E., Kral, A. G., Janssen, K., & Bierbaum, G. (2020). Experimental taphonomy of fish—Role of elevated pressure, salinity and pH. *Scientific Reports*, 10(1), 7839. <https://doi.org/10.1038/s41598-020-64651-8>

Gaines, R. R. (2014). Burgess Shale-type Preservation and its Distribution in Space and Time. *The Paleontological Society Papers*, 20, 123-146. <https://doi.org/10.1017/S1089332600002837>

Gonzalez, C. R. M., & Bradley, B. P. (1995). Are there salinity stress proteins? *Marine Environmental Research*, 39(1), 205-208. [https://doi.org/10.1016/0141-1136\(94\)00034-M](https://doi.org/10.1016/0141-1136(94)00034-M)

Hof, C. H. J., & Briggs, D. E. G. (1997). Decay and mineralization of mantis shrimps (Stomatopoda; Crustacea); a key to their fossil record. *PALAIOS*, 12(5), 420-438. [https://doi.org/10.1043/0883-1351\(1997\)012<0420:DAMOMS>2.0.CO;2](https://doi.org/10.1043/0883-1351(1997)012<0420:DAMOMS>2.0.CO;2)

Iniesto, M., Buscalioni, Á. D., Carmen Guerrero, M., Benzerara, K., Moreira, D., & López-Archilla, A. I. (2016). Involvement of microbial mats in early fossilization by decay delay and formation of impressions and replicas of vertebrates and invertebrates. *Scientific Reports*, 6(1), Article 1. <https://doi.org/10.1038/srep25716>

Iniesto, M., Laguna, C., Florín, M., Guerrero, M. C., Chicote, A., Buscalioni, A. D., & LÓPEZ-ARCHILLA, A. I. (2015). The impact of microbial mats and their microenvironmental conditions in early decay of fish. *PALAIOS*, 30(11), 792-801. <https://doi.org/10.2110/palo.2014.086>

Iniesto, M., Lopez-Archilla, A. I., Fregenal-Martínez, M., Buscalioni, A. D., & Guerrero, M. C. (2013). Involvement of microbial mats in delayed decay : An experimental essay on fish preservation. *PALAIOS*, 28(1), 56-66. <https://doi.org/10.2110/palo.2011.p11-099r>

Iniesto, M., Villalba, I., Buscalioni, A. D., Guerrero, M. C., & López-Archilla, A. I. (2017). The Effect Of microbial Mats In The Decay Of Anurans With Implications For Understanding Taphonomic Processes In The Fossil Record. *Scientific Reports*, 7(1), Article 1. <https://doi.org/10.1038/srep45160>

Jung, Y. J. (2002). Characterization of Bioactive Compounds Obtained from Starfishes. *Doctoral Dissertation, M. Sc., Dissertation. Graduate School of Changwon National University.*

Murdock, D. J., Gabbott, S. E., Mayer, G., & Purnell, M. A. (2014). Decay of velvet worms (Onychophora), and bias in the fossil record of lobopodians. *BMC Evolutionary Biology*, 14(1), 222. <https://doi.org/10.1186/s12862-014-0222-z>

Muscente, A. D., Schiffbauer, J. D., Broce, J., Laflamme, M., O'Donnell, K., Boag, T. H., Meyer, M., Hawkins, A. D., Huntley, J. W., McNamara, M., MacKenzie, L. A., Stanley, G. D., Hinman, N. W., Hofmann, M. H., & Xiao, S. (2017). Exceptionally preserved fossil assemblages through geologic time and space. *Gondwana Research*, 48, 164-188. <https://doi.org/10.1016/j.gr.2017.04.020>

Naimark, E., Kalinina, M., Shokurov, A., Boeva, N., Markov, A., & Zaytseva, L. (2016). Decaying in different clays : Implications for soft-tissue preservation. *Palaeontology*, 59(4), 583-595. <https://doi.org/10.1111/pala.12246>

Namaei Kohal, M., Esmaeili Fereidouni, A., Firouzbaksh, F., & Hayati, I. (2018). Effects of dietary incorporation of *Arthrospira* (Spirulina) platensis meal on growth, survival, body composition, and reproductive performance of red cherry shrimp *Neocaridina davidi* (Crustacea, Atyidae) over successive spawnings. *Journal of Applied Phycology*, 30(1), 431-443. <https://doi.org/10.1007/s10811-017-1220-5>

Reichart, O., Szakmár, K., Jozwiak, A., Felföldi, J., & Baranyai, L. (2007). Redox potential measurement as a rapid method for microbiological testing and its validation for coliform determination. *International Journal of Food Microbiology*, 114(2), 143-148. <https://doi.org/10.1016/j.ijfoodmicro.2006.08.016>

Saleh, F., Antcliff, J. B., Birolini, E., Candela, Y., Corthésy, N., Daley, A. C., Dupichaud, C., Gibert, C., Guenser, P., Laibl, L., Lefebvre, B., Michel, S., & Potin, G. J.-M. (2024). Highly resolved taphonomic variations within the Early Ordovician Fezouata Biota. *Scientific Reports*, 14(1), 20807. <https://doi.org/10.1038/s41598-024-71622-w>

Saleh, F., Bath-Enright, O. G., Daley, A. C., Lefebvre, B., Pittet, B., Vite, A., Ma, X., Mángano, M. G., Buatois, L. A., & Antcliff, J. B. (2021). A novel tool to untangle the ecology and fossil preservation knot in exceptionally preserved biotas. *Earth and Planetary Science Letters*, 569, 117061. <https://doi.org/10.1016/j.epsl.2021.117061>

Saleh, F., Clements, T., Perrier, V., Daley, A. C., & Antcliff, J. B. (2023). Variations in preservation of exceptional fossils within concretions. *Swiss Journal of Palaeontology*, 142(1), 20. <https://doi.org/10.1186/s13358-023-00284-4>

Saleh, F., Ma, X., Guenser, P., Mángano, M. G., Buatois, L. A., & Antcliff, J. B. (2022). Probability-based preservational variations within the early Cambrian Chengjiang biota (China). *PeerJ*, 10, e13869. <https://doi.org/10.7717/peerj.13869>

Saleh, F., Pittet, B., Perrillat, J.-P., & Lefebvre, B. (2019). Orbital control on exceptional fossil preservation. *Geology*, 47(2), 103-106. <https://doi.org/10.1130/G45598.1>

Saleh, F., Pittet, B., Sansjofre, P., Guériaux, P., Lalonde, S., Perrillat, J.-P., Vidal, M., Lucas, V., El Hariri, K., Kouraiss, K., & Lefebvre, B. (2020). Taphonomic pathway of exceptionally preserved fossils in the Lower Ordovician of Morocco. *Geobios*, 60, 99-115. <https://doi.org/10.1016/j.geobios.2020.04.001>

Saleh, F., Qi, C., Buatois, L. A., Mángano, M. G., Paz, M., Vaucher, R., Zheng, Q., Hou, X.-G., Gabbott, S. E., & Ma, X. (2022). The Chengjiang Biota inhabited a deltaic environment. *Nature Communications*, 13(1), 1569. <https://doi.org/10.1038/s41467-022-29246-z>

Saleh, F., Vaucher, R., Antcliff, J. B., Daley, A. C., El Hariri, K., Kouraiss, K., Lefebvre, B., Martin, E. L. O., Perrillat, J.-P., Sansjofre, P., Vidal, M., & Pittet, B. (2021). Insights into soft-part preservation from the Early Ordovician Fezouata Biota. *Earth-Science Reviews*, 213, 103464. <https://doi.org/10.1016/j.earscirev.2020.103464>

Sansom, R. S. (2016). Preservation and phylogeny of Cambrian ecdysozoans tested by experimental decay of Priapulid. *Scientific Reports*, 6(1), Article 1. <https://doi.org/10.1038/srep32817>

Sansom, R. S., Gabbott, S. E., & Purnell, M. A. (2010a). Decay of vertebrate characters in hagfish and lamprey (Cyclostomata) and the implications for the vertebrate fossil record. *Proceedings of the Royal Society B: Biological Sciences*, 278(1709), 1150-1157. <https://doi.org/10.1098/rspb.2010.1641>

Sansom, R. S., Gabbott, S. E., & Purnell, M. A. (2010b). Non-random decay of chordate characters causes bias in fossil interpretation. *Nature*, 463(7282), 797-800. <https://doi.org/10.1038/nature08745>

Sansom, R. S., Gabbott, S. E., & Purnell, M. A. (2013). Atlas of vertebrate decay : A visual and taphonomic guide to fossil interpretation. *Palaeontology*, 56(3), 457-474. <https://doi.org/10.1111/pala.12037>

Suliman, M., Tinay, A., Elkhalfifa, A. E., Babiker, E., & Elkhailil, E. (2006). Solubility as Influenced by pH and NaCl Concentration and Functional Properties of Lentil Proteins Isolate. *Pakistan Journal of Nutrition*, 5. <https://doi.org/10.3923/pjn.2006.589.593>

Tomas, A. L., Sganga, D. E., Marciano, A., & López Greco, L. S. (2020). Effect of diets on carotenoid content, body coloration, biochemical composition and spermatophore quality in the “red cherry” shrimp *Neocaridina davidi* (Caridea, Atyidae). *Aquaculture Nutrition*, 26(4), 1198-1210. <https://doi.org/10.1111/anu.13076>

Trevino, S. R., Scholtz, J. M., & Pace, C. N. (2008). Measuring and Increasing Protein Solubility. *Journal of Pharmaceutical Sciences*, 97(10), 4155-4166. <https://doi.org/10.1002/jps.21327>

Whitaker, A. F., Schiffbauer, J. D., Briggs, D. E. G., Leibach, W. W., & Kimmig, J. (2022). Preservation and diagenesis of soft-bodied fossils and the occurrence of phosphate-associated rare earth elements in the Cambrian (Wuliuan) Spence Shale Lagerstätte. *Palaeogeography, Palaeoclimatology, Palaeoecology*, 592, 110909. <https://doi.org/10.1016/j.palaeo.2022.110909>

Wijnker, J. J., Koop, G., & Lipman, L. J. A. (2006). Antimicrobial properties of salt (NaCl) used for the preservation of natural casings. *Food Microbiology*, 23(7), 657-662. <https://doi.org/10.1016/j.fm.2005.11.004>

Wilson, J., Wilson, L., & Patey, I. (2014). The influence of individual clay minerals on formation damage of reservoir sandstones : A critical review with some new insights. *Clay Minerals*, 49. <https://doi.org/10.1180/claymin.2014.049.2.02>

ROUND 1 REVIEWER 1 ATTACHMENT:

Reviewer: Julie Bartley, Gustavus Adolphus College

KEY RESULTS—This experimental taphonomy paper examines redox changes in four decaying invertebrates over a period of seven days and concludes that larger, protein-rich organisms reach reducing conditions more rapidly than smaller, lipid-rich organisms. The paper further proposes that these results suggest that inter-taxon differences could lead to different taphonomic pathways in organisms entering the taphonomic window under identical initial conditions.

VALIDITY—The experimental results are intriguing and provide a potentially fruitful set of hypotheses for future work, both in experimental taphonomy and in the fossil record. The predictions are certainly testable. However, the conclusions are not strongly supported by the data available in these experiments. The conclusion states that larger organisms (relative to the water they decompose in) exert a greater influence on redox conditions than smaller organisms (Fig 2A); however, the order of normalized redox impact (greatest to least) is

shrimp>**snail**>**starfish**>planarian

while the mass order (highest mass to lowest) is

shrimp>**starfish**>**snail**>planarian

In addition, the paper concludes that protein content (relative to lipids and carbohydrates) is also an important driver of redox change, such that organisms with higher protein content utilize oxygen faster during decay. The order of protein content (highest to lowest) is (Fig 2C)

Planarian>**starfish**>**snail**>shrimp

while the mass-adjusted redox impact (greatest to least) is

Planarian>**snail**>**starfish**>shrimp

Thus, although the planarian and shrimp follow the proposed pattern, the two intermediate organisms, snails and starfish, do not. Additionally, while the masses of the snail and starfish samples are quite close, their normalized ORP paths are quite different from one another.

Based on this reading of the results, it does not appear that the primary conclusions of the paper are well-supported by the data. That said, the results are suggestive. Specifically, when the carcass mass:water ratio is high, it is easier for redox potential to change quickly. However, this is a matter of simple stoichiometry – oxygen becomes a limiting reactant under these conditions. Natural systems are not closed in this way, though they may be diffusion-limited and thus act somewhat like a closed system. It would not be unreasonable to propose that a larger carcass might experience diffusion-limited local redox change more quickly than a smaller carcass. It's possible, perhaps, that the authors could probe that relationship by examining the individual carcasses by mass to ascertain whether they show a pattern related to size. Such an analysis would likely have limited statistical support, but a suggestive pattern might be interpretable in light of the overall findings.

SIGNIFICANCE—The authors correctly point out that taphonomic variations, even among organisms buried under similar conditions (e.g., the same bedding plane), are very difficult to explain. This approach and these preliminary results represent a useful way to pursue this question. I think it is likely that the conclusion of this paper is basically correct; my concern is that the data do not demand this explanation and I do not think the uncertainty is fully addressed in the article.

DATA AND METHODOLOGY—

The methodology suffers from a crucial flaw, and that is that the experimental systems are “closed” to varying degrees. The average shrimp carcass is more than 25x larger than the average planarian, and both were decomposed in the same volume of water (and thus the same effective O₂ concentration). It is likely that the planarian chamber remained an open system with respect to the carcass throughout the experiment, while the shrimp carcass exhibited closed-system behavior for nearly the entire experiment duration. Unfortunately, the only remedy I see for this flaw is to repeat the experiment with either similar biomass in each chamber or oxygen availability scaled to carcass size (i.e., use more water for larger carcass sizes). When reviewing

manuscripts, I try very hard not to send the authors back to the bench, but here I'm not seeing much of an alternative. My suggestion above (examining individual carcasses by mass) might mitigate the issue, but likely would not solve it.

I have a few smaller questions about methodology that are probably easily addressed by the authors:

- Does the reported mass include skeletons? If not, please specify that and specify how you determined skeletal mass separately from soft tissue mass. If the skeleton is included in the reported mass, it would be even harder for me to support the conclusion of the mass-redox relationship, as the skeletons aren't likely participating in the redox reactions.
- Line 111 indicates that proportions of proteins, lipids, and carbohydrates were obtained from the literature. Were the taxa used in the experiment the same taxa analyzed in the literature sources? I don't think it's likely to matter very much, but this should be clarified.
- Was the head space in each vial filled with fully oxygenated air? It is worth specifying that condition as well.

ANALYTICAL APPROACH—

Statistics is not my area of expertise, but there are a few small items that would clarify the analyses:

- Figure S1 should specify the variance across what variable? I am thinking it is ORP, but please be explicit.
- Specify the threshold for significance. It looks like differences are significant at $p < 0.05$, but that should be stated.
- I don't understand the utility of the analysis of daily change in ORP values. First of all, reversing the sign on the "normalized ORP daily difference variable" such that this variable is positive when the observed ORP value goes *down* is unnecessary and confusing (I spent much too long trying to figure that out, even though it is stated in the figure caption). Sign of the variable aside, this analysis does not seem to contribute anything to the overall conclusions.

In Figure 1 and figures 2A, 2B, the figure caption needs to explain what the line represents. It appears to be smoothed, rather than a mean line. And presumably, the dots are the individual measurements, but it would be good to be explicit about that as well.

SUGGESTED IMPROVEMENTS—I think I've summarized possible improvements to the manuscript in the categories above. As I said, I really hesitate to tell authors to perform additional experiments, so perhaps analyzing the mass-ORP relationships a little more closely would provide additional support for the hypothesis.

CLARITY AND CONTEXT—

Overall, the manuscript is clearly written. I have a few suggestions organized by line number:

L69 Change censor to **sensor**

L70 the variance **in ORP** [if indeed it is the variance in ORP that's being analyzed]

L105 Does the body mass include the skeleton?

L125 Change sample size to **number of samples** [to avoid confusion with body mass]

L163 change In this sense to **in this situation**

L166 change frees to **releases**

L188-191 I don't think you can draw that conclusion quite so strongly, because of the closed-system behavior likely experienced by the shrimp. Also, snails have less protein than starfish, so why would they experience a more marked drop in ORP, according to the proposed explanation?

L192 change proximate to **approximate**

L192 remove , near the end of this line

Fig 2 change proximate on the y-axis to **approximate**

L229-230 This is only true in a closed nutrient system. On the seafloor, we would expect water to be circulating over the tops of these carcasses, so in this sense, the experiment is not a good model for pre-burial taphonomy; it is a better model for early post-burial, where diffusion limits the supply of O₂ to the carcass.

L262 change organic matters to **organic matter types**

References

- In several references, genus and species names are not in italics. See, for example, ref #s 6, 15, 18, 35, 39, 45.
- Ref #26: *pap* needs to be capitalized
- Most references use the full journal title, but a few use an abbreviation (cf. 10, 22, 26, 35). Probably, they should all be the same.